# Scoping Review of Yoga in Schools: Mental Health and Cognitive Outcomes in Both Neurotypical and Neurodiverse Youth Populations

**DOI:** 10.3390/children9060849

**Published:** 2022-06-08

**Authors:** Niamh Hart, Samantha Fawkner, Ailsa Niven, Josie N. Booth

**Affiliations:** 1Physical Activity for Health Research Centre (PAHRC), Institute for Sport, Physical Education and Health Sciences, Moray House School of Education and Sport, University of Edinburgh, Edinburgh EH8 8AQ, UK; niamh.hart@ed.ac.uk (N.H.); ailsa.niven@ed.ac.uk (A.N.); 2Institute for Education, Community and Society, Moray House School of Education and Sport, University of Edinburgh, Edinburgh EH8 8AQ, UK; josie.booth@ed.ac.uk

**Keywords:** yoga, schools, physical activity, mental health, cognition

## Abstract

Yoga is used widely as a therapeutic tool for physical and mental well-being. The mind-body activity could be an inclusive and cost-effective intervention used within schools to help tackle the mental health crisis in youth populations. To date, research has focused on mainly neurotypical youth populations. However, greater acknowledgement of the impact for neurodiverse youth populations is warranted. Therefore, the aim of this scoping review is to understand what is known about the relationships between yoga in schools and mental health and cognition in neurotypical and neurodiverse youth populations. Methods: This review followed O’Malley and Arskey’s methodological framework. A comprehensive database search using fundamental keywords and index terms was conducted. Screening was carried out in Covidence^TM^ software. Results: Substantial evidence to support the use of school-based yoga programmes for the improvement of anxiety, self-concept, resilience, depression, self-esteem, subjective and psychological well-being, executive function, inhibition, working memory, attention and academic performance in neurotypical populations was found. Evidence to support school-based yoga programmes in neurodiverse populations with improvements in self-concept, subjective well-being, executive function, academic performance and attention was also found. Conclusions: The findings support the provision of yoga in schools to improve mental health and cognition whilst also creating clear pathways for future research and school-based yoga intervention development.

## 1. Introduction

In UK youth populations (YP) (Throughout this article, the term ‘youth population’ is used to describe school-aged (4–18 years) children and young people.), 1 in 8 present with at least one mental health difficulty (e.g., anxiety, depression) with rising prevalence rates from 9.7% in 1999 to 11.2% in 2017 [1]. However, the experienced trauma associated with pandemics can cause short-term and long-term effects on YP’s mental health [2]. Therefore, it is likely that these percentages have increased following the restrictions imposed worldwide following the emergence of COVID-19. A recent systematic review [3] describing mental health in YP during the first year of the COVID-19 pandemic found that most studies included in the review observed a general trend of worsening mental health and increases in depressive and anxiety symptoms in participants.

A population that is at higher risk for mental health comorbidities is neurodiverse youth [4] due to high levels of social, emotional and education associated impairments and lifelong persistence [5]. Neurodiverse youth are those who require Additional Support for Learning (ASL) needs that are different from those of the same age to ensure they benefit from education, whether during early learning, school or preparation for life after school [6]. Amongst YP requiring ASL, it is estimated that around 2–5% of school-aged youth in the UK have Attention Deficit Hyperactivity Disorder (ADHD) [7], and around 1 in 57 are on the Autistic spectrum [8].

Many treatments and therapies for mental health are either expensive, ineffective or not readily available to all [9]. Additionally, medication often presents with adverse side effects [10]. A potential low-risk complementary therapy for mental health in YP is physical activity (PA) [11,12]. To experience the multiple physical and mental health benefits of PA, the WHO recommend that YP participate in an average of at least 60 min of moderate to vigorous PA daily [13]. PA is associated with positive outcomes on cognitive function and academic attainment and is beneficial for a range of neurological and mental health difficulties in YP [11,12,13]. One potential PA intervention for YP is yoga.

Yoga is an ancient, mind-body practice that incorporates breathing regulation, physical postures, and meditation [14]. It is a low-cost, low-risk and inclusive mode of PA [9] that has been shown to have physical and mental health benefits in YP. There is review level evidence that supports the role of yoga in neurotypical YP in improving fitness and lean body mass [15,16] and promising effects on psychosocial well-being, stress [16,17], anxiety [18,19], resilience and self-regulation [15]. However, to date, reviews on yoga in YP have been focused on the physical and mental health effects in predominantly neurotypical YP. These reviews potentially have limited relevance for neurodiverse YP, with just one systematic review including neurotypical YP [19]. Serwacki and Cook-Cottone [19] noted effects supporting the provision of yoga in reducing stress in YP with autism as well as improvements in attention in those with severe educational problems. However, only 25% of their included study population were neurodiverse; therefore, it is difficult to confidently support the provision of yoga for youth who require ASL from these findings.

Schools provide an ideal location to deliver PA interventions with a broad reach to neurotypical and neurodiverse YP of all ages [20]. It is therefore critical to look specifically at school-based interventions to understand if yoga as part of the school day is feasible and effective. However, to the authors’ knowledge, only two reviews [17,19] have examined yoga interventions explicitly in the school setting. Whilst therapeutic benefits in psychosocial well-being (anxiety, stress and negative affect) [17] and positive effects on academic outcomes (IQ scoring and contribution in the classroom) [19] were reported, it is important to note these studies’ limitations. Both reviews are now over 7 and 10 years old, respectively, and as yoga is being utilised much more widely in the school environment [21], the research field has developed and needs updating. Furthermore, there is a clear under-representation of synthesised literature on neurodiverse YP and yoga in schools, who, ironically, may benefit most from yoga as part of their school day [22].

Therefore, the aim of this scoping review was to map the relationship between yoga in schools and mental health and cognitive outcomes in neurotypical and neurodiverse YP and identify any potential differences in outcomes between the two populations. Our proposed research question is broad to map out key concepts in the research area and to identify research priorities to support the provision of yoga in schools. The findings will enable stakeholders in education and yoga to make informed choices about utilising yoga within the school day.

Research Question: what is known about the relationships between yoga in schools and mental health and cognition in neurotypical and neurodiverse youth populations?

This scoping review has the following three objectives:Scope out the different mental health and cognitive functioning outcomes for neurotypical youth populations receiving a yoga intervention as part of the school day;Scope out the different mental health and cognitive functioning outcomes for neurodiverse youth populations receiving a yoga intervention as part of the school day;To explore the differences in the outcomes for neurotypical and neurodiverse youth populations.

## 2. Materials and Methods

This study employed a systematic search of the literature in the form of a scoping review. Unlike systematic reviews, a scoping review allows for a broader approach to searching key concepts and types of sources within a research area [23]. When exploring a complex research area that has not been comprehensively reviewed before, the scoping review is useful to determine and present the existence and range of the evidence available in a systematic and meaningful way [23]. The methodological framework for this scoping review is based on the 5-Stage Process by O’Malley and Arksey [23], along with extra recommendations for each stage by Levac et al. [24] and followed the Preferred Reporting Items FOR Systematic Reviews and Meta-Analyses (PRISMA) extension for scoping reviews checklist [25] (Appendix A). The framework discusses the need for the scoping review to be an iterative process, with the use of expert discussion throughout [23]. Terminology in the research area of neurodiversity area can be problematic, and there is a lack of consensus on modern definitions [26]; however, for the purposes of this research, we have used the definitions listed in Table 1. The protocol [15] for this review is registered with The Open Science Framework.

Stage 1: Identify the research question

A broad research question was proposed following an initial literature search: What is known about the relationships between yoga in schools and mental health and cognition in neurotypical and neurodiverse youth populations?

Stage 2: Identify relevant studies

A preliminary literature search indicated that studies in this area vary in study design, mode of yoga used in the intervention and in the population targeted and outcomes measured. The following eligibility criteria were decided on after multiple discussions within the research team and with the experience from the initial literature review and upon analysing other systematic reviews on yoga in youth populations. The decision was made to include a variety of study designs and studies that included any type of yoga or yoga-inspired intervention and a mixture of outcome variables, where it was possible to disentangle the mental health and cognitive outcomes. Only peer-reviewed literature was included to increase the likelihood of including higher-quality information.

Inclusion criteria included: research from any geographical location, English language, school setting, both sexes, school-aged young people 4–18 years old, neurotypical and neurodiverse youth populations, all forms of yoga (e.g., hatha, ashtanga, yoga-inspired), and peer-reviewed sources of information. The review included primary research studies with pre-post outcomes measured, and qualitative research, evaluating mental health (MH) and/or cognitive functioning outcome variables (key definitions found in Table 2 below). These MH and cognitive outcomes were selected and constructed from Lubans et al. [28] “Conceptual model for the effects of physical activity on mental health outcomes in children and adolescents. ADHD, attention-deficit/hyperactivity disorder”. Studies that assessed MH and cognitive outcomes in addition to other outcomes such as physical health or prosocial behaviour were also included where it was possible to disentangle the findings.

Exclusion criteria: non-English-speaking language, opinion pieces, magazines or newspaper articles, dissertations or books, reviews, research articles that do not scientifically investigate measures of mental health or cognitive functioning and studies that include additional therapies used in combination with yoga interventions (e.g., yoga and Thai Chi) where it is not possible to disentangle.

### Review Search Strategy


*Step 1: An initial search*


An initial search was conducted on Google Scholar to become familiar with the evidence base using the words yoga AND schools. The first 100 studies were reviewed, and 24 studies were identified as relevant with information extracted from the year, name of the study, reference, study design, methods, population, outcomes measured, and outcomes found. The initial search reviewed all outcome variables, including physical and mental health, cognitive outcomes, and prosocial behaviour; however, after discussions with the research team, it was deemed most relevant to focus on mental health and cognitive outcomes. The individual reference lists of the 24 studies were also examined to identify further relevant studies.


*Step 2: Identify keywords and index terms*


The initial literature search enabled the author to identify fundamental index terms and keywords for the main searches. To ensure inclusivity, “yoga” and “school*” were the main terms used in the database searches.

Secondary search terms (adolescen*, teen* mental health, depression, anxiety, self-esteem, self-concept, psychological stress, psychological well-being, subjective well-being, resilience, social-isolation, loneliness, cogniti*, executive function*, inhibition, shifting, academic attainment, academic achievement, IQ, and cognitive outcome*, well-being, cognitive function*) were combined with the Boolean operator OR and then combined with the yoga and school terms using AND to facilitate the recovery of relevant studies (The search strategies and terms can be found in Appendix A). Searches were conducted on 13 April 2021 and updated on 7 April 2022.


*Step 3: Searching electronic databases*


The database search was designed to be as comprehensive as possible. After reviewing previous reviews in the field [16,19] and a consultation with an academic information specialist, a selection of databases (MEDLINE, PsycINFO, SPORTDiscus, CINAHL) were selected to reach saturation of searches with education, psychology and physical activity as the main areas of interest. The databases were searched for titles with “yoga” and “school*” and secondary search terms.


*Step 4: Citation and reference lists*


The reference and citation list of relevant studies were also searched to identify further relevant studies not picked up by the database searches. Authors of significant primary studies, scoping reviews and systematic reviews were contacted to locate any additional important sources of information that could not be retrieved through the online searches.

Stage 3: Study selection

Study selection required 2 steps; 

(1)All identified studies were uploaded to Covidence software, where duplicates were automatically removed on upload.(2)Titles and abstracts were screened by N.H. with 100% double screened between the members of the research team. The entire research team (N.H., S.F., A.N. and J.B.) double screened 30% of the full-text review as a quality assurance measure. This increased confidence for one author (N.H.) to continue the subsequent 70% full-text review. If a paper could not be retrieved during the study selection process, the author(s) were contacted to request a copy. However, if the paper was not recovered, the study was excluded.

Stage 4: Charting the data

Data were extracted, interpreted and synthesised as per best practice reported by Peters et al. [41]. A data extraction form was utilised to ensure all relevant data were extracted from each study. The data extraction form followed the headings: year of publication, reference, author(s), location, title, synopsis, study design, no. of participants, participant info (including gender, age, neurodiverse/neurotypical), type of yoga, components of yoga, intervention details, measurements used, outcomes measured in relation to RQ, effects found and implications.

N.H. carried out the extraction process whilst meeting with the research team at regular intervals to ensure the approach was consistent with answering the scoping review research question.

Stage 5: Collating, summarising and reporting the results

The findings are presented in two ways (1) table format, a table with study characteristics and (2) thematically, descriptive analysis presenting key themes and concepts relevant to the research question.

## 3. Results

### 3.1. Characteristics of All Studies

The study selection flowchart is presented in Figure 1. A total of 695 references were identified for screening. Following the removal of duplicates, 561 studies were screened by title and abstract, and subsequently, 217 full-text papers were screened for eligibility. A total of 59 studies met the inclusion criteria and were included for data extraction. The 59 studies took place in the following locations: U.S (*n* = 32), India (*n* = 16), Germany (*n* = 2), UK (*n* = 2), New Zealand (*n* = 2), Israel (*n* = 1), Canada (*n* = 1), Columbia (*n* = 1), Sri Lanka (*n* = 1) and Tunisia (*n* = 1). Only 10% (*n* = 6) of studies were conducted before 2010, presenting a substantial and recent increase in the number of related studies. A total of 41% of studies (*n* = 24) were conducted in the last 5 years.

### 3.2. Neurodiverse Youth Populations

Table 3 details the findings relating to 11 studies, 9 experimental [42,43,44,45,46,47,48,49,50] and 2 qualitative [51,52], that included neurodiverse youth populations. Yoga classes varied in length from 10–120 min, from classes taken daily to every other week, integrated into classroom teaching or before, during or after school hours. Neurodiverse youth were described as displaying physical impairments [42], learning and educational difficulties [43,48], functional impairments [50], ADHD [45], varying degrees of autism [52] and emotional and behavioural disorders (EBD) [47,48,50]. Two studies suggested from their methods or results that some of their sample were neurodiverse but did not record details or diagnoses [44,51]. Neurodiverse youth populations totalled 19% of the study samples. Of these 11 studies, 18% [49,51] reported on both cognitive and mental health outcomes, 64% [43,44,45,46,47,48,50] assessed just cognitive outcomes and 18% [42,52] on mental health alone.

#### 3.2.1. Neurodiverse Youth and Mental Health Outcomes

##### Self-Concept

One pre-post study [42] and one qualitative study [51] found improvements in measures of self-concept within neurodiverse youth populations on completion of a school-based yoga program (SBYP). Case-Smith et al. [51] investigated students’ perceptions of an SBYP designed as a preventive intervention to reduce stress and improve behaviour in students at risk for learning difficulties. Supporting a positive self-concept (“I feel like I’m in a good place”) was a theme that emerged, with participants describing positive feelings about themselves and feelings of self-affirmation. For example, one participant explained that yoga “gives you power, and you can be strong” [51]. Berwal and Gahlawat’s [42] post-intervention results showed improvement in all dimensions of self-concept, including academic, social, temperamental, educational, moral and intellectual dimensions of self-concept.

##### Anxiety

One pre-post study [49] assessed outcome measures of anxiety with neurodiverse youth following an SBYP. Participants with emotional and behavioural disorders (EBD) showed a significant increase in state anxiety following the yoga programme; however, the authors reported mitigating external factors that may explain this unexpected increase in anxiety.

##### Subjective Well-Being

Steiner et al. [49] included outcome measures of subjective well-being within their pre-post intervention. Results found that parents (72%) reported positive changes in their children (with EBD), including increased happiness, upon completion of an SBYP. Participants with varying degrees of autism from a qualitative study [52] reported improvements in positive affect through reflective journaling. Participants expressed phrases such as “I breathed in my body. I feel well” and “I feel good. I did deep breathing, relaxed me”.

#### 3.2.2. Neurodiverse Youth and Cognitive Health Outcomes

##### Attention

One quasi-experimental [46] and one qualitative [51] study assessed outcomes of attention. Powell et al. [46] found greater improvements in a control group (CG) on “concentration/attention skills in the classroom” compared to a yoga group. Conversely, a positive theme of attention emerged in Case-Smith et al. [51], with one participant expressing that “the yoga program helps us like when we are doing a project or something, it helps us and it makes us focus about what we are about to do and it helps us” [51].

##### Executive Function

A pre-post study of youth with Emotional Behavioural Disorders (EBD) [49] assessed executive function outcomes. Post-intervention, teachers reported significant improvement on the Internalizing Problems Composite, Behavioural Symptoms Index and Adaptive Skills Composite and a trend toward significant improvement on the School Problems and Externalizing Problems composites assessed by the Behaviour Assessment Scale for Children, Second Edition Parent Rating Scale—Child (BASC-2 PRS-C).

##### Academic Performance

One pre-post [45], one quasi-experimental study [50] and one RCT [48] explored academic performance in neurodiverse youth. Mehta et al. [45] reported improvement in performance impairment scores in a large number of participants, especially for those rated in the poor academic and social performance categories. A total of 57 of the 63 (90.5%) children had some form of improvement in their performance impairment scores following the yoga programme, and more than half (55.5%) of participants improved to the normal range scores of ‘no performance impairment’ reported by teachers. Smith et al. [48] reported there were no significant changes to grades; however, some changes to standardised test scores could be interpreted as showing a benefit following the SBYP. Uma et al. [50] assessed the use of daily yoga practice on IQ (intelligence quotient) scores in participants with functional impairment. Pre-post scores of IQ and mental age revealed significant improvement for those in the yoga treatment versus the control group. Of the yoga group, 89% significantly improved their general mental ability scores compared to 57% in the control. Additionally, 68% of the yoga group significantly improved their IQ capacities, as opposed to only 41% in the control group.

### 3.3. Neurotypical Youth

Table 4 details the findings relating to 48 studies, 37 experimental [53,54,55,56,57,58,59,60,61,62,63,64,65,66,67,68,69,70,71,72,73,74,75,76,77,78,79,80,81,82,83,84,85,86,87,88,89], 7 qualitative [90,91,92,93,94,95,96] and 4 mixed [97,98,99,100] methods that included neurotypical youth populations. Yoga classes varied in length from 10–120 min. Yoga was practiced daily to every other week, integrated into classroom teaching or before, during or after school hours. Neurotypical youth populations totalled 81% of the study samples. Of these studies, 40% [58,64,65,66,71,76,77,79,81,82,84,85,93,94,95,96,97,98,100] reported on both cognitive and mental health outcomes, 31% [56,57,59,60,69,72,75,78,80,87,88,89,91,92,99] assessed just cognitive outcomes and 29% [53,54,55,61,62,63,67,70,71,73,74,83,90] on mental health alone.

#### 3.3.1. Neurotypical Youth and Mental Health Outcomes

##### Resilience

Statistically significant improvements in resilience outcomes were noted in two studies [62,98]. Further, breathing and meditation techniques were discussed qualitatively [98], with participants expressing feelings of self-control and calming effects, resulting in reduced stress and greater overall well-being. Teachers also reported that the students were increasingly tolerant in class [98]. The use of didactic discussions around self-awareness and dealing with stressors was used by Felver et al. [62] and Noggle et al. [74]; however, the studies found contrasting results. Noggle et al. [74] did not find any statistically significant resilience outcomes between the yoga and control group.

##### Self-Esteem

Significant improvements in total [55,67,83,85], general [54,84], social [54,66] and parental [84] self-esteem (SE) were found following an SBYP. However, Gulati et al. [66] also found adverse effects, with general SE scores decreasing significantly. The author noted that this could be due to a lack of parental support, as the participants were attending a residential school. Furthermore, both Bridges and Madlem [55] and Telles et al. [84] found that SE increased in both control and yoga groups. Qualitatively, two studies reported positive effects of yoga on SE [95,96]. Interviews yielded positive perceptions of students’ self-esteem from their school teachers [95]. Focus groups also provided the following quotation from a student’s experience of yoga over an academic year, “Taught me that I am capable of doing more than I thought I could” [96].

##### Self-Concept

Two experimental studies [71,79] reported self-concept outcomes following an SBYP. Kundu and Pramanik [73] investigated self-concept outcomes of two individual yoga elements, breath work (pranayama) and postures (asanas). Results revealed postures, breath work, and a combination of the two significantly improved the measures of self-concept of school-going children. Although small, the group with both breathing techniques and postures experienced the most improvement in self-concept measurements. However, Richter et al. [79] reported a significant increase in physical self-concept of speed in a physical skill group (CG) compared to a yoga group.

##### Depression

A total of eight experimental studies explored yoga and effects on depression in school-based interventions [58,63,65,73,74,76,86,97]. Students participating in SBYP benefitted from decreases in depression [63,97] with four studies showing decreases relative to control groups [73,74,76,86]. One study [65] recruited a sample of youth who were self-reporting high levels of depressive symptoms, with results indicating that baseline depressive symptoms may distinguish the youth who benefit most from the program. Specifically, for youth reporting low to medium levels of baseline depressive symptoms, the intervention reduced Involuntary Engagement and Impulsive Action stress responses compared to the control group. However, the same effects were not apparent between intervention and control groups in those who reported high levels of depressive symptoms at baseline. Adverse effects of depression were reported in Butzer et al. [58] when the entire sample, regardless of yoga or control (PE), reported significant increases in depression during the intervention duration.

##### Anxiety

Ten studies explored anxiety outcomes following a school-based yoga programme [63,71,73,74,76,77,81,83,85,86]. Anxiety was significantly reduced in five studies [63,83], with three significantly reduced compared with control groups [74,76,81,86]. One study [81] found that 63% of the participants in the yoga program had elevated anxiety scores at the start of the program. After 8 weeks of 10 min of daily yoga in the classroom, only 40% had elevated anxiety scores. An additional study [71] revealed that anxiety was significantly improved in the Asana Pranayama (postures and breathing) combined group, followed by a breathing only group and a posture only group. Two qualitative studies [96,100] reported improvements in anxiety outcomes following an SBYP. One quote by a participant reflected that the use of breathing techniques taught during yoga allowed them to reduce their anxiety, “The inhale and exhale helped my anxiety. I would do it when I became anxious during the day” [96]. Anecdotal evidence by teachers suggested that children who were the most stressed and anxious had released their anxiety and no longer showed signs of stress in the classroom following yoga [100].

However, three studies also showed ambiguous results. Tersonde de Paleville and Immekus [85] did not find any significant reductions in anxiety following an SBYP. Quach et al. [77] found that all three treatments, yoga group, meditation group, and control group, had reductions in anxiety outcomes. They concluded such decreases in anxiety could not be solely attributed to the “active ingredients” of the school-based interventions as the waitlist control also had reductions in anxiety outcomes. An additional study [73] found particularly mixed findings in anxiety outcomes following an SBYP in four different schools. Participants in the yoga program at School A reported significant decreases in anxiety compared with those in the comparison condition. However, these significant effects were not reported within the other three intervention groups in School C, B or D. Interestingly, the participants in the yoga program at School C reported significant increases in anxiety compared to students in the comparison condition, with the author attributing this to differences in school environments across the four schools.

##### Psychological and Subjective Well-being

Nine experimental studies assessed either subjective or psychological well-being outcomes as part of an SBYP and found significant improvements including quality of life [53], mood [61,74], positive affect [70,98], negative affect [70,74], happiness [71], satisfaction [71], and non-significant improvements in positive affect [63,67] and negative affect [63,97]. However, a number of studies failed to find any significant improvements in negative affect [64], positive affect [64,74], or well-being measures [68,82]. Furthermore, Haden et al. [67] and Sarkissian et al. [98] reported increases in negative affect following an SBYP.

Five qualitative studies that explored subjective and psychological well-being reported positive effects [51,90,93,94,100], particularly in positive affect, with a participant from Case-Smith et al. [51] expressing, “When I do yoga it made me happy”, and another participant from Rashedi et al. [94] stating that the breathing techniques made them feel good with the quote, “I love breathing because it feels like ice cream”. Another participant from Laxman [93] stated that the yoga postures allowed them to experience “peace within”. Psychological well-being was also expressed with yoga making one participant feel “healthy and strong” [94].

#### 3.3.2. Neurotypical Cognitive Outcomes

##### Inhibition

Four experimental studies considered inhibition outcomes [58,65,78,97]. Improvements in inhibition scores after an SBYP were found in a mixed-methods [97] and a quasi-experimental [78] study in involuntary stress responses of ‘rumination’ and ‘intrusive thoughts’ [78,97], and whilst it was not significant, a positive direction was reported for measurements in ‘impulsive action scores’ [97]. A quasi-experimental study [65] assessing an SBYP on impulsive action scores recruited a sample with elevated depressive symptoms at baseline. It was reported that those with lower levels of depressive symptoms reported significantly lower impulsive action post-intervention. Interestingly, this effect was not significant in youth reporting medium to high levels of depressive symptoms. An RCT [58] reported similar findings on inhibition and links to depression. Participants reported higher levels of depression over time in a 6-month trial. Females in the yoga group reported a statistically significant increase in ‘lack of premeditation’ scores between time 1 versus time 2, then a significant decrease from time 2 versus time 3, whereas males in the yoga group did not report any changes in ‘lack of premeditation’.

##### Attention

Existing evidence suggests that yoga interventions may increase attention of youth populations in school, with all 15 studies with an attention measurement reporting improvements [44,51,57,60,66,69,75,80,84,85,92,93,94,95,100]. Experimental evidence from trial designs found significant improvements on attention compared to control [57,60,66,69,80]. Conversely, one study [85] found improvements but not at a statistically significant level. A comparative study of yoga and physical education (PE) found that both yoga and PE resulted in significant improvements in attention [84]. Similarly, Pandit and Satish [75] found that all treatments, a yoga group, a health intervention group and a time-lagged group, all had enhanced attention scores at 3 and 6 months from baseline, which the author described as natural maturation. Qualitative research found that both teachers and young people reported positive effects from school-based yoga programmes on attention [45,52,94,95,101]. Authors also reported increased attention with “alert states” [94] and increased “focus” [95].

##### Working Memory

Two RCTs assessed working memory outcomes following an SBYP [77,88]. Results from Quach et al. [77] found that working memory capacity was significantly increased in a meditation treatment but not in the yoga or control group. In contrast, one study [88] found significant improvement in one of four memory tests in the yoga group, compared to no improvements in the control group.

##### Executive Function

Five studies found significant improvements in executive function outcomes following an SBYP [82], including four studies comparing results with control groups [72,81,87,89]. However, an RCT [68] and one quasi-experimental study [79] found no significant associations with executive function in either yoga or control groups.

##### Academic Performance

Experimental studies found significant improvements in academic performance as a result of an SBYP [57,68,81,84], whereas two studies [59,85] found improvements in academic performance but not at a statistically significant level. Contrary to this, Frank et al.’s [64] results showed no significant differences in academic achievements compared to the control groups. Another study [56] did not find direct improvement in GPA (grade point average) scores; however, they suggested that an SBYP may have had a preventive effect by reducing declines in GPA over time, especially compared to the control who revealed a steeper downward trend in GPA. Four qualitative studies reported improvements in academic performance following a school-based yoga programme [92,96,97,100]. Butzer et al. [91] reported at least 25% of their participants increasing academic achievements, with one student quoting, “I didn’t notice it at first but I noticed since I started yoga in term 2. In term 1 my science grade was like a C but last term in science I got a B and this term I’m pretty sure I have an A average right now. And I sort of feel like that’s because of yoga, because I have science immediately after yoga” [91]. Students may experience increased academic performance through the calming effects of breathing that they learn through yoga, as suggested by a participant from Wang and Hagins [96], “I learned a lot about the brain and nervous system. We have a lot of tests and when I have yoga in the morning it is very calming. It is a way to calm down before the test”. One study [92] reported little direct impact on academic performance; however, participants from the same study expressed improvements in their attitudes towards school and decreases in academic-related stress. In a mixed-methods study [98], participants attributed their success in achieving higher grades to the SBYP.

##### Summary of Neurodiverse and Neurotypical Findings

The third objective of this scoping review was to explore the differences in the outcomes for neurotypical and neurodiverse youth populations. Table 5 synthesises the findings for each population by favourable outcomes. Less than 50% of studies in an outcome finding a positive effect is illustrated as red (weak evidence), 51–75% of studies in an outcome finding a positive effect as amber (moderate evidence) and more than 76% is illustrated as green (strong evidence). For example, 7 out of 10 studies exploring anxiety outcomes in neurotypical populations found favourable outcomes, and this is represented as moderate evidence in amber. However, only one study explored anxiety in neurodiverse populations and failed to find a positive outcome, so this is represented as weak evidence in red. Grey indicates areas with no evidence. For example, there were no studies found exploring self-esteem outcomes in neurodiverse populations.

## 4. Discussion

This scoping review aimed to map out the relationships between yoga in schools and mental health and cognition in neurodiverse and neurotypical youth populations. From the results of this study, we identified 59 studies from all over the world using randomised controlled trials, quasi-experimental, pre-post, within-groups, mixed methods, longitudinal and qualitative designs. We found that school-based yoga programmes (SBYP) are incredibly heterogeneous in nature, and the literature is multi-dimensional. Where consistent literature does exist, there is clear positive evidence supporting the beneficial effects of yoga in schools and some clear gaps where future research should be targeted. There was strong evidence to suggest that SBYP improve outcomes of resilience, self-esteem, self-concept, depression, anxiety, psychological well-being, inhibition, attention, working memory, executive function and academic performance in neurotypical youth populations. There was considerably less literature exploring yoga in schools and neurodiverse youth populations. However, improvements in self-concept, subjective well-being, academic performance and executive functioning were reported following an SBYP.

### 4.1. Comparison with Literature and Plausible Explanations for Findings

Our principal findings are consistent with those from the most recent systematic review [16] that yoga in youth populations can lead to mental and cognitive health outcomes; however, Miller et al. [16] did not focus entirely on educational settings or on neurodiverse youth. Our review has also added to the literature by including a further seven studies to the synthesis of literature around yoga in schools with neurodiverse youth populations since Serwacki and Cook-Cottone’s [19] systematic review that included this population. Subsequently, this review’s additional evidence supports the provision of yoga for neurodiverse youth populations by improving self-concept [42,53], attention [44,49,51], subjective and psychological well-being [49,51,52], academic performance [45,48], executive function [49] and depression [49]. Therefore, this review builds on existing evidence by providing an updated broad review of yoga in schools and mental health and cognitive outcomes in both neurotypical and neurodiverse youth populations.

### 4.2. Mental Health Outcomes and Neurodiverse Populations

There was some positive evidence to support the use of yoga in neurodiverse populations for the improvement of self-concept and subjective well-being. However, the findings from this scoping review demonstrate a gap in the literature supporting yoga as a tool for promoting mental health in youth populations that require additional support for learning (ASL). There was no evidence of improvements in resilience, self-esteem or depression outcomes in neurodiverse youth. Adverse effects of anxiety outcomes in youth with emotional behaviour disorders (EBD) [49] were also reported. Yet, conversely, the same study [49] had anecdotal evidence to suggest that parents reported their children to be happier as a result of a yoga programme. Youth populations that require ASL are more likely to experience mental illness [4] due to high levels of social, emotional and education associated impairments [5]; however, this review cannot confidently recommend the use of yoga-based interventions for these populations due to lack of evidence in the area. This indicates a clear pathway for further research.

### 4.3. Mental Health Outcomes and Neurotypical Populations

There was clear evidence to support the use of yoga in schools in neurotypical populations to improve self-esteem, depression, and subjective and psychological well-being. There was some evidence to indicate that resilience, self-concept and anxiety are improved upon completion of an SBYP. These findings are consistent with those from past reviews [15,16,17,19]. This supports Hartley and Henderson’s [9] call for nationwide use of SBYP to improve the mental well-being of youth populations.

Some adverse effects were noted in depression outcomes in one study with neurotypical youth [58]; however, this was seen across both the yoga and control group, which may suggest that there was another external factor impacting this outcome that the study could not account for. Another adverse effect of an SBYP was discovered in negative affect, which was also noted by Miller et al. [16]. One postulation for this could be that exposure to a new discipline (yoga) could perhaps increase negative affect in the short term [67]. Again, further research is required to substantiate any positive or adverse effects found in mental health outcomes.

### 4.4. Differences in the Mental Health Outcomes for Neurotypical and Neurodiverse Populations

There is a stark contrast in the evidence available for neurodiverse populations compared with neurotypical populations and mental health outcomes following an SBYP. Where there was consistent evidence in neurotypical populations and anxiety, depression, self-esteem, self-concept, psychological and subjective well-being and resilience outcomes, there was either none or minimal for neurodiverse. This difference is represented in Table 5. For example, there is clear evidence to support the improvement of depression outcomes in neurotypical youth, with 7 out of 8 studies (88%) reporting positive results. However, there was no evidence available for depression outcomes in neurodiverse youth populations.

### 4.5. Cognitive Outcomes and Neurodiverse Populations

Whilst only three studies explored academic performance outcomes in neurodiverse youth, 100% found improvement. Interestingly, one study [45] noted that participants diagnosed with ADHD and categorised in low academic and social performance categories reported the most improvements in performance impairment scores, with more than half of the sample improved to normal range scores of ‘no performance impairment’. This finding suggests that yoga could be potentially very valuable for those who require ASL. One study [49] found improvements in scores of executive function in youth with emotional and behavioural disorders. Another study [51] found improvements in attention outcomes in a sample of youth, of whom some required ASL. However, it is impossible for this scoping review to definitively say that those who reported the most improvement in attention from the SBYP [51] were the participants with ASL needs. It is also important to note that there was no representation for inhibition, shifting and working memory outcomes in neurodiverse youth populations.

### 4.6. Cognitive Outcomes and Neurotypical Populations

The findings of this review support the use of SBYP for cognitive outcomes in neurotypical populations, as reflected in previous reviews [15,17,19]. Improvements in attention, inhibition, working memory, executive function and academic performance were found. These findings also mirror evidence from a recent systematic review on cognitive outcomes in youth populations following physical activity (PA) [101]. Bidzan-Bluma and Lipowska [101] found that increased PA led to improvements in executive function and working memory. However, the present review reports mixed findings on the effects of yoga in schools on working memory and inhibition, making it difficult to make any definitive conclusions from these findings. One study [65] suggested that less severe depressive symptoms may increase improvements in inhibition. In addition, an increase in working memory was reported in a meditation group, not the yoga group [77], whereas Verma et al. [88] noted that yoga improved working memory in a yoga group rather than the physical skill control group. These findings need further investigation to understand which type of PA is most effective in improving certain cognitive outcomes.

### 4.7. Differences in the Cognitive Outcomes for Neurotypical and Neurodiverse Populations

PA improves circulation leading to a better oxygen supply to the brain, which is said to aid the maturation of motor development and nervous impulses [101]. This is particularly important in childhood when the maturation of cognitive functioning is at its highest [102] and into ‘sensitive periods’ of development in adolescence [103]. However, as observed in Table 5, there is a clear difference in the volume of evidence supporting SBYP and cognitive outcomes in neurodiverse compared to neurotypical populations. There is strong positive evidence to support SBYP and attention (100% of 15 studies) and academic performance (92% of 13 studies) in neurotypical populations. Whilst 100% (3 out of 3) of studies recommended SBYP for improvements in academic performance in neurodiverse populations, there was 77% less representation of this population in the literature. In contrast, only 50% (1 out of 2 studies) found improvements in attention in neurodiverse populations. Moreover, there were no studies in either population to support SBYP and the cognitive function of shifting.

### 4.8. Strengths and Limitations

The strengths of this scoping review are that it followed a rigorous framework [23], was conducted in concordance with PRISMA guidance and followed a published protocol [104] in line with the Open Science Framework. It included a diverse range of study designs allowing a more comprehensive overview of the literature. It is the first scoping review of yoga and schools with neurotypical and neurodiverse youth populations. It provides a synthesis of current literature on mental health and cognitive outcomes, and lastly, it maps out patterns of positive findings and potential research gaps.

This review included mental health and cognitive outcomes as proposed by Lubans et al. [28] conceptual model for the effects of PA on mental health outcomes in children and adolescents. We did not explore the potential underlying psychosocial and behavioural mechanisms responsible for any positive or negative effects of yoga in schools on mental and cognitive health outcomes or explore why and under what conditions these mental and cognitive outcomes occur. However understanding this may increase SBYP’s effectiveness in the future [28]. This is also a clear pathway to explore future interventions, perhaps with the use of an established conceptual framework created for the development, delivery and evaluation of such interventions.

In the previously published protocol, the inclusion criteria stated the use of studies that utilise the four main components of yoga, physical postures [27] and movement, breath work, relaxation techniques and meditation/mindfulness practices [15]; however, to avoid the exclusion of highly relatable evidence it was decided in the screening stages to include studies that incorporated at least two main yoga components instead of three. The study protocol also stated the use of pre-post peer-reviewed evidence, but again, upon screening, a variety of important qualitative literature was discovered, and therefore we included all peer-reviewed empirical and non-empirical evidence, including qualitative studies. The value of a scoping review is that it does not require ‘strict limitations on search terms, identification of relevant studies, or study selection at the outset’ [23], and we were able to make adjustments as we became more familiar with the evidence base.

Furthermore, due to the nature of the scoping review, we did not appraise the quality of evidence. Rather, we have been able to scope out all evidence in the area and map out any potential gaps. Another limitation of this review is that due to time and resource constraints, only 30% of full-text double-screening occurred; therefore, there is a possibility that some relevant information was missed. Nevertheless, it could be argued that one review is unable to completely acquire all available evidence, and we are confident that this review presents an excellent overview of yoga in schools and the relationship with mental health and cognitive outcomes in neurotypical and neurodiverse youth populations.

### 4.9. Future Research and Implications

Schools provide an ideal location to promote PA among youth populations with broad reach and numerous opportunities throughout the school week, including before, after and even during school class time [20]. The findings of this study support the use of yoga as an effective school-based intervention, with substantially more benefits outweighing any minimal negative effects reported. However, there are a few important aspects to consider for future research in this area and for the development of SBYP.

Our findings show a substantial increasing interest in yoga and schools, with almost half of the studies identified published in the last five years. However, since Serwacki and Cook-Cottone [19], only seven additional peer-reviewed studies have been completed in neurodiverse youth populations, with only one (4%) [54] completed in the last five years when 41% of our review’s studies were published. The evidence base is substantially growing, but there is a clear gap in clinical and sub-clinical populations. Future research in this area should aim to explore the role of yoga in schools for YP with neurodiversity. Some positive evidence emerged from the findings to suggest that yoga may be beneficial to improve the mental and cognitive well-being of neurodiverse youth populations; however, rigorous methods, such as RCTs, are needed.

Furthermore, whilst appraising study quality was not the aim of this scoping review, it is encouraging that 69% of the studies included used a control group. Future studies should aim to use rigorous methods such as randomised controlled trials with matched controls to reduce bias. Additionally, only two studies were conducted in the United Kingdom, and therefore it may be difficult to generalise any findings from these findings to a UK population when education and curriculum differ from one country to another.

## 5. Conclusions

School-based yoga programmes have the potential to improve mental health and cognition in neurotypical youth populations. There is strong positive evidence to support the use of SBYP for the improvement of depression, self-esteem, subjective and psychological well-being, attention and academic performance and moderate evidence to support anxiety, self-concept, resilience, executive function, inhibition and working memory in neurotypical populations. Whilst there was a stark contrast in the volume of evidence present for the use of SBYP in neurodiverse populations, there was some strong positive evidence to support improvement in self-concept, subjective well-being, executive function and academic performance and moderate evidence for outcomes in attention. This scoping review highlights the clear deficiencies in the research field of yoga and neurodiverse populations. We recommend that future research aims to bridge this gap by examining the potential of SBYPs that are inclusive to all students.

## Figures and Tables

**Figure 1 children-09-00849-f001:**
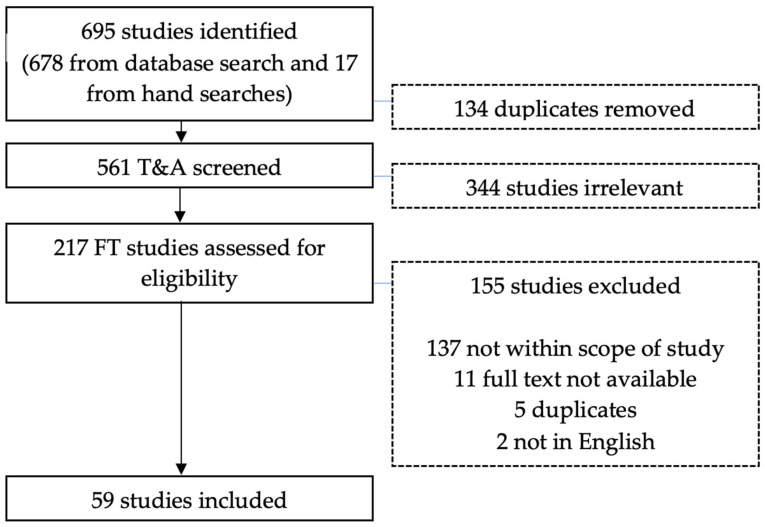
Study selection flowchart.

**Table 1 children-09-00849-t001:** Key Definitions.

Terminology	Definition Used
Yoga in Schools	Yoga interventions either before, during or after school. Studies that utilise at least two of the four main components of yoga: physical postures [27] and movement, breathwork, relaxation techniques and meditation/mindfulness practices [15].
Neurotypical	Those who are developing like others of the same age and are not receiving additional or different support.
Neurodiverse	Those who require additional support that is different from that received by children of the same age to ensure they benefit from education, whether early learning, school or preparation for life after school [6].

**Table 2 children-09-00849-t002:** Mental Health and Cognitive Outcome Definitions.

Terminology	Definition Used
**Mental Health Outcomes**	Adapted from Biddle, Ciaccioni, Thomas and Vergeer [11] and Lubans et al. [28].
Anxiety	Activation of the automatic nervous system with distressing thoughts and/or feelings of tension, agitation, excessive worry or apprehension about certain events (such as environment, social, academic, occupational) [29].
Depression	Extended periods of low mood and loss of interest or pleasure in generally all activities [29].
Self-esteem	An individual’s evaluation of their own worth [30].
Self-concept	An individual’s awareness or beliefs regarding their qualities and limitations both globally and in specific subdomains (e.g., academic, physical, social) [31].
Psychological Well-being	Psychological well-being links with autonomy, environmental mastery, personal growth, positive relations with others, purpose in life and self-acceptance. This is often referred to as eudemonic well-being [29].
Subjective-Well-being	Subjective well-being is defined as a person’s cognitive and affective evaluations of their life. SWB is closely aligned with the construct of happiness [32].
Resilience	A personality characteristic that moderates the negative effects of stress and enables individuals to successfully cope with challenges and misfortune [33].
**Cognitive Outcomes**	Adapted from Diamond [34] and Welsch, Alliott, Kelly, Fawkner, Booth, Niven [35].
Executive Function	The cognitive processes that are used to carry out new or difficult tasks. These processes include inhibition, working memory, shifting and attention [34].
Inhibition	The control of attention, behaviour, thoughts and emotions by overriding internal tendencies or external distractions [34].
Working Memory	The of holding information in the mind and working with it mentally [36], e.g., thinking of a response while listening to a conversation.
Shifting	The flexibility to adjust to changed demands or priorities [34].
Attention	The ability to focus on information for several seconds (interrelated with working memory) [34].
Academic Performance	Academic performance broadly refers to the communicative (oral, reading, writing), mathematical, science, social science and thinking skills and competencies that enable a child to succeed in school and society. Because these forms of achievement are difficult to assess, most researchers have relied on a narrower definition that is largely limited to outcomes on standardised achievement tests [37].
IQ	Intelligence quotient [38] refers to mental age (MA) expressed as a ratio of chronological age (CA) multiplied by 100. For IQ to remain stable, MA must increase with CA over time [39].
**Neurodevelopmental Impairments**	The American Psychiatric Association [29] define neurodevelopmental disorders that present themselves at the onset of the developmental stage through personal, social, academic and occupational impairments.
ADHD	Attention-Deficit/Hyperactivity Disorder is defined by the American Psychiatric Association [29] as a consistent pattern of inattention, impulsivity and/or hyperactivity.
ASD	Autism Spectrum Disorder [38] is a complex developmental disorder that involves persistent challenges in social interaction, restricted and repetitive behaviours and speech and non-verbal communication [29]. The severity and effects of ASD differ between each individual.
Learning Difficulties	Identified as a particular type of “unexpected” low achievement and distinguished from types where low achievement is expected due to emotional disturbance, social or cultural disadvantage or inadequate instruction [40].

**Table 3 children-09-00849-t003:** Study Characteristics of Included Studies with Neurodiverse Youth.

Author & Year	Methods	Control Group (CG)	Participants	Intervention	Outcomes	Effects Found
**Berwal and Gahlawat, 2013** [42]	Pre-post	NO	*n* = 15 14–18 years No gender details Visually impaired students drawn from a special school for the blind in India.	Ashtanga and theory yoga—Postures, breathing, meditation and theory classes on the importance of yoga 30 days daily for 60 min.	Self-concept	Improvement in all dimensions of self-concept, including academic, intellectual and social.
**Case-Smith, Shupe Sines and Klatt, 2010** [51]	Qualitative	NO	*n* = 21 7.4 years (mean) F = 12/M = 9 3 participants received additional support for learning.	Hatha yoga—Postures, breathing, meditation and relaxation + 15 min activities at end for self-concept (e.g., making an Olympic medal to wear or a collage of people who loved them) 45 min × 1 a week for 8 weeks. In addition, 4 days a week, the teacher of the class led 15 min yoga in the classroom.	Self-concept, attention	Three themes emerged: (1) feeling calm/focused, (2) controlling own behaviour, (3) supporting a positive self-concept suggesting that yoga programs with at-risk children may enhance well-being and positive self-concept.
**Hopkins and Hopkins, 1979** [43]	Within-groups	YES—general PE activities	*n* = 34 6–11 years No gender details Not labelled but exhibit severe educational problems	Yoga—Postures, breathing and meditation 2 conditions—15 min for 8 or 22 sessions (duration not detailed)	Attention	Both treatments were followed by more efficient completion of the criterion task than CG.
**Klatt, Harpster, Browne, White and Case-Smith, 2013** [44]	Mixed methods	NO	*n* = 41 8.54 years (mean + 0.55) F = 25/M = 16 The teachers describe some of the participants as having ADHD symptoms.	Mindfulness-based Intervention (MIL)—Postures, meditation, background music, art activities with weekly overarching theme, e.g., health, support, success 45 min × 1 a week for a total of 8 weeks conducted during the school day in the classroom	Attention	Assessment of student behaviour on both ADHD index and in cognitive/inattentive behaviour showed decreases in disruptive behaviours.
**Laxman, 2022** [52]	Qualitative	NO	*n* = 3 15–18 years Participants were conveniently recruited as they had learning and intellectual disabilities and varying degrees of autism.	Goldberg’s Creative Relaxation Programme designed for young people with autism—postures, breathing and relaxation 15 min × 1 week for 5 weeks in the classroom	Subjective well-being	Improvements in positive affect were noted by some students.
**Mehta, Mehta, Mehta, Shah, Motiwala, Vardhan, … and Mehta, 2011** [45]	Pre-post	NO	*n* = 63 6–11 years F = 26/M = 44 Participants were recruited as they had been diagnosed previously with ADHD using the Vanderbilt questionnaire as well as having a neurodevelopmental assessment by a neurodevelopmental paediatrician.	Multimodal programme that incorporates yoga as well as behavioural play therapy—Postures, breathing and meditation, behavioural play 60 min × 2 a week over 6 weeks during the school day	Academic performance	More than 50% of the children improved their academic and behavioural performance.
**Powell and Potter, 2010** [47]	Pre-post	NO	*n* = 21 11–15 years M = 21 Of the 21 pupils, 6 were diagnosed with an EBD alone, 2 were diagnosed with EBD and ADHD, 1 was diagnosed with ADHD alone, 1 was reported by teachers to have ADHD and epilepsy, and 1 child had a diagnosis of global delay. The remaining 9 pupils had mild to severe learning disabilities. Six pupils were in receipt of additional help.	Hatha yoga and self/peer massage—Postures, massage, breathing, meditation, relaxation and visualisation 60 min × 12 classes over 2 school terms in a room in the school	Attention	Positive change was noted by teachers of pupils’ reduced hyperactivity. No statistically significant changes, but there were trends toward improvements in attention span and eye contact with teachers.
**Powell, Gilchrist and Stapley, 2008** [46]	Quasi-experimental	YES—regular additional support as normal	*n* = 107 8–11 years F = 48/M = 59 Participants exhibited emotional, behavioural and learning difficulties.	Self-Discovery Programme—Postures, breathing, relaxation, communication and massage 45 min × 1 a week over 12 weeks	Attention	The yoga group had significant improvements in ‘contribution in the classroom’; however, there were greater trends towards improvement in attention in CG.
**Smith, Connington, McQuillin and Crowder Bierman, 2014** [48]	Randomised controlled trial	YES—CG attended Healthy Eats: a non-physical activity	*n* = 77 9.38 years (+0.97) F = 41/M = 36 16 of the 77 students were categorised as ‘special education’.	‘YogaKidz’ Curriculum—Postures, breathing, relaxation and didactic themes 40 min × 2 a week for 28 weeks	Academic performance	Better growth in reading scores for the yoga group as opposed to decline in scores for CG.
**Steiner, Sidhu, Pop, Frenette and Perrin, 2013** [49]	Pre-post	NO	*n* = 37 8–11 years (10.4 mean age) F = 15/M = 22 9.8% had attention problems (ADHD), 9.8% depression/bi-polar, 19.5% anxiety/OCD, 24.4% behaviour problems, 24.4% autism spectrum disorder, 7.3% neurological problems and 58.5% school problems including speech and language, reading.	‘Yoga Ed’ Protocol—Postures, breathing, relaxation, social component of partner/group exercises, imagery and meditation 60 min × 2 a week for 3.5 months during school hours	Anxiety, attention, psychological and subjective well-being, executive function.	Teachers reported improved attention in class, adaptive skills and reduced depressive symptoms. Children did report increased anxiety.
**Uma, Nagendra, Nagarathna, Vaidehi and Seethalakshmi, 1989** [50]	Quasi-experimental	YES—no treatment	*n* = 90 6–16 years F = 32/M = 58 Students were selected based on mild, moderate and severe functional impairment such as IQ and adaptive skills. Amongst those who were included in the study, 12 pairs (pairs refer to control and treatment balanced) belonged to the mild degree (IQ 50–70), 17 pairs belonged to the moderate degree (IQ 35–50), and 16 pairs belonged to the severe degree (IQ 20–34).	Yoga—Postures, breathing, meditation 60 min × 5 days a week for 10 months	IQ	IQ scores improved significantly in the yoga group compared to the CG.

**Table 4 children-09-00849-t004:** Study Characteristics of Included Studies with Neurotypical Youth.

Author & Year	Methods	Control Group (CG)	Participants	Intervention	Outcomes	Effects Found
**Bazzano, Anderson, Hylton and Gustat, 2018** [53]	Randomised controlled trial	YES—’care-as-usual’ (CG)	*n* = 52 8–9 years Positive for symptoms of anxiety using the SCARED scale F = 25/M = 27 Typically developing.	‘Yoga Ed’—Postures, breathing and meditation 40 min × 10 sessions for 8 weeks held in a classroom in the mornings	Subjective well-being	The yoga group demonstrated significantly greater improvements in psychosocial and emotional quality of life compared with CG.
**Berezowski, Gilham and Robinson, 2017** [90]	Qualitative	NO	*n* = 3 16–18 years No gender details	‘Yoga 11 curriculum’—Postures, breathing, meditation, philosophy and life skills Participants completed the Yoga 11 course as part of their Physical Education. Duration unknown.	Subjective well-being	Participants generally expressed how yoga made them feel happier (affect).
**Bhardwaj and Agrawal, 2013** [54]	Randomised controlled trial	YES—no treatment but free to complete homework or reading	*n* = 44 10–12 years (mean 11.27) F = 18/M = 26	Yoga—Postures, breathing and relaxation and OM chanting 35 min × 6 days a week for one month during school time	Self-esteem	The yoga group demonstrated a significant rise in the level of total self-esteem, general self-esteem and social self-esteem compared to CG. No significant changes found in the academic or parental self-esteem outcomes.
**Bridges and Madlem, 2007** [55]	Quasi-experimental	YES—traditional PE	*n* = 53 13–14 years F = 24/M = 29	Yoga—no info 40 min × 2 a week for 16 weeks	Self-esteem	Self-esteem increased in both the experimental and CG with no significant differences between the two groups.
**Butzer, Van Over, Noggle Taylor and Khalsa, 2015** [56]	Randomised controlled trial	YES—traditional PE	*n* = 95 14–17 years F = 55/M = 40	‘Kripalu Yoga’—Postures, breathing, game/activity and meditation and relaxation 35–40 min × 2/3 times a week for 12 weeks	Academic performance	Both groups exhibited a decline in GPA over the school year. However, CG exhibited a significantly greater decline in GPA over time than the yoga group.
**Butzer, Day, Potts, Ryan, Coulombe, Davies,… and Khalsa, 2015** [57]	Pre-post	NO	*n* = 36 6–8 years F = 16/M = 30	‘Yoga 4 Classrooms’ programme—Postures, breathing, meditation and relaxation + themed discussion at start class to promote self-inquiry and reflection. 30 min weekly class over 10 weeks. Taught during regular class time	Attention and academic performance	Teacher reported significant improvements in attention span, ability to concentrate on work, ability to stay on task and academic performance.
**Butzer, LoRusso, Windsor, Riley, Frame, Khalsa and Conboy, 2017** [91]	Qualitative	YES—traditional PE	*n* = 16, 13.27 years (+0.42) F = 8/M = 8	Kripalu Yoga in the Schools (KYIS)—Postures, breathing, didactic/experiential content and relaxation 35 min × 1 or 2 a week for 6 months	Academic performance	Participants mentioned using breathing techniques to help prepare for tests, while other students mentioned improvements in academic performance, with 25% showing improvements through Valence analysis.
**Butzer, LoRusso, Shin and Khalsa, 2017** [58]	Randomised controlled trial	YES- traditional PE	*n* = 209 12.64 years (+0.33) F = 132/M = 77	Kripalu Yoga in the Schools (KYIS)—Postures, breathing, didactic/experiential content and relaxation 35 min × 1 or 2 a week for 6 months	Depression and inhibition	The entire sample (yoga and CG) reported significant increases in depression.
**Conboy, Noggle, Frey, Kudesia and Khalsa, 2013** [92]	Qualitative	NO	*n* = 28 15 years F = 17/M = 11	Kripalu Yoga (KYIS)/classical Hatha yoga style—Postures, breathing exercises, deep relaxation and meditation techniques 30 min for 12 weeks	Attention and academic performance	Little direct effect on grades. Participants noted that yoga helped to relieve academic stress and improve attitudes towards school. Students used the breathing techniques taught in the yoga classes to prepare for exams. Other students noted that yoga during the day improved their ability to focus and concentrate.
**Dabre, Pingale, Tejrao and Humbad, 2011** [59]	Quasi-experimental	YES—other school activities	*n* = 154 11–13 years No gender details	Yoga—Postures, breathing and meditation Once a day as part of school’s morning routine, 6 days a week for 10 months	Academic performance	There was a rise in the school results of yoga group, whereas no difference between the CG and school results.
**Ehud, An and Avshalom, 2010** [60]	Pre-post	NO	*n* = 122 8–12 years No gender details	Iyengar- Postures and breathing 13 yoga sessions conducted over 4 months incorporated into school schedule	Attention	There was a significant improvement of attention.
**Felver, Butzer, Olson, Smith and Khalsa, 2014** [61]	Within-groups	YES—traditional PE	*n* = 47 14–16 years F = 52%/M = 48%	Kripalu Yoga in Schools (KYIS)—Postures, breathing, meditation and relaxation 35 min × 5 days for 3 weeks	Subjective well-being	Immediate improvements in mood and affect were noted following both yoga and CG; however, the yoga class had a larger effect than traditional PE.
**Felver, Razza, Morton, Clawson and Mannion, 2020** [62]	Quasi-experimental	YES—art or music class	*n* = 23 12.1 years F = 12/M = 11	Kripalu Yoga in Schools (KYIS) + Normal PE programming—Postures, breathing, relaxation and didactic theme 45 min × 7 days a week over 7 weeks (*n* = 33)	Resilience	Participants in the yoga group demonstrated significant improvements in resilience over time, whereas scores in the CG did not significantly change.
**Frank, Bose and Schrobenhauser-Clonan, 2014** [63]	Quasi-experimental	NO	*n* = 49 14–18 years F = 27/M = 22	TLS (Transformative Life Skills)-A manualised universal classroom-based programme—Postures, breathing and meditation. Sessions integrated into the classroom 30 min × 3–4 days per week during the first semester of the school year.	Subjective well-being, anxiety, depression	No statistically significant differences in measures of positive affect and negative affect were found. However, the general direction of scores was in predicted direction. Significant improvements were found in measures of student anxiety and depression.
**Frank, Kohler, Peal and Bose, 2017** [64]	Randomised controlled trial	YES—’business as usual’	*n* = 159 11–15 years F = 74/M = 81	TLS (Transformative Life Skills)-A manualised classroom-based programme—Postures, breathing and meditation. Sessions integrated into the classroom 30 min × 3–4 days per week during the first semester of the school year.	Academic performance, subjective well-being	No differences between groups were noted on measures of positive, negative affect or grades. However, as compared to the CG, students in the yoga intervention had improved levels of school engagement.
**Gould, Dariotis, Mendelson and Greenberg, 2012** [65]	Quasi-experimental	YES—waitlist	*n* = 97 9–11 years F = 59/38	Holistic Life Foundation (HLF)—Postures, breathing, meditation and relaxation 45 min × 4 days a week for 12 weeks. Classes were held in the school’s physical activity rooms, e.g., hall or gym, during ‘resource time’—a period in which students engage in non-academic activities	Depression, inhibition	Baseline depressive symptoms moderated both impulsive action and involuntary engagement stress responses. The yoga group reporting lower levels of baseline depressive symptoms were more likely to report decreases in impulsive action and involuntary engagement responses relative to CG.
**Gulati, Sharma, Telles and Balkrishna, 2019** [66]	Pre-post	NO	*n* = 116 10.2 (+0.6) years F = 38/M = 78	Yoga—Postures, breathing, relaxation and chanting 60 min daily × 7 days a week	Self-esteem, attention	Improvements in attention, concentration and self-esteem (social, academic, and total) were found.
**Haden, Daly and Hagins, 2014** [67]	Randomised controlled trial	YES—traditional PE	*n* = 30 10–11 years F = 13/M = 17	Ashtanga—Postures, breathing, relaxation and meditation 1.5 h × 3 times a week for 12 weeks	Subjective well-being and psychological well-being	No significant differences between groups. Negative affect increased in yoga group and decreased in CG, as well as a non-significant increase in positive affect in the yoga group.
**Hagins and Rundle, 2016** [68]	Randomised controlled trial	YES—traditional PE	*n* = 112 14–17 years F = 54/M = 58	‘Sonima Foundation Yoga curriculum’—Postures, breathing, meditation and relaxation and discussion on didactic themes 45 min × 2 a week over the academic year (*n* = 58 classes)	Academic performance, executive function, well-being	Yoga group participants had a higher mean GPA than CG. No changes to executive function were found. There was a trend of poorer well-being scores in the yoga group.
**Jarraya, Wagner, Jarraya and Engel, 2019** [69]	Randomised controlled trial	YES—PE (active CG) OR no treatment	*n* = 45 5.2 years (+0.4) F = 28/M = 17	Hatha Yoga—Jogging/jumping warm-up, breathing, yoga postures, sensory games and story 30 min × 2 a week over 12 weeks (total *n* = 24) during normal kindergarten hours	Attention	The yoga group, in comparison to both CGs, had a significant positive impact on inattention and hyperactivity.
**Kale and Kumari, 2017** [70]	Randomised controlled trial	YES—no details	*n* = 60 12–15 years M = 60	Yoga—Prayer, postures, breathing, mantra chanting, cleansing processes and relaxation OR CG (no details) 60 min × 6 days a week in school afternoon	Subjective well-being	There was a significant positive improvement in positive affect and a significant reduction in negative affect in the yoga group.
**Kundu and Pramanik, 2014** [71]	Randomised controlled trial	YES—Group D had no treatment	*n* = 120 8–10 years M = 120	Group A—postures only OR Group B—breathing only OR Group C—postures and breathing 45 min × 6 days a week for 12 weeks in the school activity hall	Self-concept, subjective well-being, anxiety	Anxiety was significantly improved in all groups but showed better effects in Group C. Happiness and satisfaction were significantly improved in all groups. Self-concept was significantly improved in all groups.
**Laxman, 2021** [93]	Qualitative	NO	*n* = 6 9–14 years No gender details	Yoga—Postures, breathing and meditation and relaxation 1 × a week for 60 min in school hall building for 5 weeks (total *n* = 6)	Attention, subjective well-being	A general increase in subjective well-being during and after the yoga sessions was found. Participants also reported improved attention.
**Manjunath and Telles, 2001** [72]	Randomised controlled trial	YES—traditional PE	*n* = 20 10–13 years F = 20	Yoga—Postures, breathing, meditation, relaxation and singing devotional songs 75 min × 7 days a week in residential school for one month	Executive function	Yoga significantly improved execution time and planning time. Planning time was also improved in the CG.
**McMahon, Berger, Khalsa, Harden and Khalsa, 2021** [73]	Quasi-experimental	YES—homework/outdoor play	*n* = 118 11–14 years F = 63/Male = 55	Kundalini Yoga-based Y.O.G.A for Youth (Y4Y)—Postures, breathing, meditation, relaxation, singing and mantras. Yogic principles such as intention, action, speech etc. were taught through group discussion 2 × 40 min classes a week over 6 weeks	Depression, anxiety	Participants in the yoga group reported significant decreases in depression after one session. Yoga’s impact on depression and anxiety depended on the school setting in which they were implemented.
**Mendelson, Greenberg, Dariotis, Gould, Rhoades and Leaf, 2010** [97]	Mixed methods	YES—waitlist	*n* = 97 9–11 years F = 59/M = 38	Holistic Life Foundation (HLF) mindfulness-based yoga—Postures, breathing, meditation, relaxation and didactic discussions based on stressors and coping mechanisms 45 min × 4 days a week for 12 weeks in a gym hall during school hours	Inhibition, subjective well-being, depression	No significant group differences on measures of mood or depressive symptoms were found, although the pattern of scores was in predicted direction for mood variables. Significant differences were found in two of the five subscales of involuntary stress responses post-intervention means, including rumination and intrusive thoughts and a trend in predicted direction for impulsive action.
**Noggle, Steiner, Minami and Khalsa, 2012** [74]	Randomised controlled trial	YES—traditional PE	*n* = 51 16–18 years F = 28/M = 23	Kripalu-based yoga programme—Postures, breathing, meditation and didactic discussion on self-inquiry and emotion regulation 30 min × 2 or 3 a week over 10 weeks (*n* = 28 total)	Subjective well-being, psychological well-being, depression, anxiety, resilience	Negative affect and tension–anxiety were all positively impacted by the intervention. However, no changes were observed in positive affect or, perceived stress, positive psychological traits, resilience., or anger expression.
**Pandit and Satish, 2014** [75]	Longitudinal	YES—three conditions including yoga, non-yoga intervention and a time-lagged comparison group	*n* = 178 9–12 years No gender details	Yoga—Postures, breathing and chanting Every other week for 12 weeks/daily practice at home	Attention and executive function	Significant changes in the attention scores in all three groups over three to six months were found. Results also showed improvement in problem-solving scores across time in all three groups over three to six months.
**Pant, 2013** [76]	Randomised controlled trial	YES—school as usual	*n* = 60 16–17 years M = 60	Yoga—Postures, breathing, meditation, relaxation and chanting 60 min × daily for 6 weeks	Anxiety and depression	Reduction of examination anxiety, depression and academic stress was found in the yoga group.
**Quach, Mano and Alexander, 2016** [77]	Randomised controlled trial	YES—mindfulness meditation OR PE waitlist	*n* = 198 (only 172 accounted for in gender) 12–15 years (13.18) F = 114/M = 58	Hatha yoga—Postures, breathing, discussion 45 min × 2 × weekly for 4 weeks during PE time in either gym or hall Participants asked to complete 15–30 min daily home practice, which was logged in a diary collected once a week	Working memory, anxiety	A significant increase in working memory was found for participants in meditation group, whereas those in yoga and CG did not present significant changes. All three groups showed significantly reduced anxiety post-intervention.
**Rashedi, Wajanakunakorn and Hu, 2019** [94]	Qualitative	Part of an ongoing RCT with waitlist control	*n* = 154 4–6 years No gender details	Yoga on pre-recorded videos—Postures, breathing, relaxation and songs 10 min × 6 weekly before lunch for 8 weeks	Subjective well-being, attention	Attention and subjective well-being improved following the yoga programme.
**Reindl, Hamm, Lewis and Gellar, 2020** [95]	Qualitative	NO	*n* = 40 students/23 teachers 8–11 years No gender details	No intervention details 1-year intervention	Attention, academic performance, self-esteem	Students perceived improved focus and academic performance whilst teachers reported increased cognitive functioning and self-esteem.
**Reid and Razza, 2022** [78]	Quasi-experimental	YES—traditional PE as normal	*n* = 112 10–12 years (mean age= 10.4 years) F = 58/M = 54	Mindfulness through Movement (MTM)—postures, breathing, relaxation and mindfulness 45 min 2 × a week for 7 months	Inhibition	Participants in the YG reported lower levels of rumination and intrusive thoughts than their peers in the CG.
**Richter, Tietjens, Ziereis, Querfurth and Jansen, 2016** [79]	Quasi-experimental	YES—physical skill training	*n* = 24 6–11 years F = 12/M = 12	Yoga embedded into story—Postures connected by story, e.g., imagining a moon and doing ‘moon pose’ 45 min × 2 a week for 6 weeks. Classes were conducted in afternoon and outside of regular sports lessons and took place in a suitable room within the school.	Executive function, self-concept	No differences between groups were found in executive function and motor skills. Participants in the yoga group reported an increase in the category speed of physical self-concept compared to CG.
**Sarkissian, Trent, Huchting and Khalsa 2018** [98]	Mixed methods	NO	*n* = 30 9–14 years F = 25/M = 5	Kundalini Yoga (Your Own Greatness Affirmed; YOGA)—Postures, breathing, meditation 50 min × 1/2 times a week for 10 weeks within the schools’ regular PE time slot or after school	Subjective well-being, resilience, academic performance	Improved positive affect and resilience were found. However, a non-significant increase in negative affect was also reported.
**Saxena, Verrico, Saxena, Kurian, Alexander, Kahlon…****and Gillan, 2020** [80]	Quasi-experimental	YES—health class	*n* = 174 14–15 years F = 112/M = 62	Hatha Yoga—Postures and meditation OR health class (CG) 25 min × 2 a week for 12 weeks. Classes took place in the morning during a required health class	Attention	Compared to CG, the yoga group reduced inattention and hyperactivity. Within the yoga group, inattention and hyperactivity symptoms diminished.
**Shreve, Scott, McNeill and Washburn, 2021** [81]	Quasi-experimental	YES—those who did not participate were the CG	*n* = 71 8–10 years F = 42/M29	Adapted 10 min version of ‘Yoga for Kids’ programme to be used in schools—Postures, breathing, relaxation and meditation 10 min × 5 days a week for 8 weeks. Yoga was completed first thing in the morning before teaching	Anxiety and academic performance	63% of the yoga group presented with elevated anxiety scores at baseline and reduced to 40% post-intervention. On average, participants in the yoga group had significantly improved academics post-intervention.
**Sinha, Kumari and Ganguly 2021** [82]	Pre-post	NO	*n* = 49 12–16 years (mean 13.6 years) F = 26/M = 23	An integrated classroom yoga module (ICYM)—Postures, breathing, chanting, affirmations and meditation 12 min × 5 days a week for 1 month	Self-esteem, well-being and executive function	Significant improvements post-intervention for executive function were found with medium–large effect sizes. Significant improvements in self-esteem were found post-intervention with a small effect size. No significant differences were found post-intervention for well-being.
**Sivashankar, Surenthirakumaran, Doherty and Sathiakumar, 2022** [99]	Mixed methods	YES—normal ‘keep fit’ routine of dancing for 20 min	*n* = 1084 13–14 years F = 549/M = 535	Yoga—Postures, breathing and meditation 20 min × 4 days a week for 6 months	Academic performance	Students noted an increase in their academic performance as a benefit of the yoga programme, with one student explaining, “I can quickly complete my homework easily”.
**Stueck and Gloeckner, 2005** [83]	Quasi-experimental	YES—no treatment	*n* = 48 11–12 years Pupils had tested for abnormal examination anxiety	‘TorweY-C’—Postures, breathing, relaxation and another element such as massage, sensory exercises, meditation, imagery techniques and interactive activities 60 min × 15 total classes	Anxiety	Immediately after intervention, anxiety was reduced and remained stable until 3 months post-intervention.
**Telles, Singh, Bhardwaj, Kumar and Balkrishna, 2013** [84]	Randomised controlled trial	YES—traditional PE	*n* = 98 8–13 years (mean 10.5 years + 1.3) F = 38/M = 60	Yoga—Postures, breathing, relaxation and chanting 45 min × 5 days a week for 3 months during school hours.	Self-esteem, academic performance, attention	Both groups showed increases in word scores, colour scores and colour-word scores. The yoga group showed an increase in total, general and parental self-esteem. Both groups showed improvements in academic performance and attention.
**Tersonde Paleville and Immekus, 2020** [85]	Randomised controlled trial	YES—physical activity program called Minds in Motion (MiM)	*n* = 44 5–11 years (mean 7.2 years) F = 21/M = 27	‘Cosmic Kids Yoga Programme’—Postures, meditation and yoga games/activities	Academic performance, anxiety, attention	Although no statistical differences were found between CG and yoga groups across measures, both groups had increased academic skills after the intervention.
**Thomas and Centeio, 2020** [100]	Mixed methods	YES—no treatment	*n* = 40 8 years F = 18/M = 22	‘Yoga Calm’—Postures, breathing, relaxation, mindfulness practice and social-emotional learning 20 min × 2 a week for 10 weeks plus shorter intervals of yoga poses, breathing and relaxation through the school day	Attention, anxiety, well-being	Participants expressed feeling ‘happier’ since yoga programme. The school teacher reported that children who were the most stressed and anxious had reduced anxiety and stress. Others became more concentrated and focused.
**Velásquez, López, Quiñonez and Paba, 2015** [86]	Randomised controlled trial	YES—waitlist	*n* = 125 10–15 years No gender details	Developed around Satyananda Yoga—Postures, breathing, meditation and relaxation 120 min × 24 over 12 weeks in school	Depression, anxiety	At baseline, the yoga group had statistically significantly higher levels of depression compared to CG. Yoga group reported a decrease in their anxiety and depression levels, while CG experienced an increase in these indicators.
**Verma and Shete, 2020** [87]	Randomised controlled trial	YES—no treatment	*n* = 66 11–15 years F = 18/M = 48	‘Kaivalydhama Yoga’—Postures, breathing, chanting 60 min × 6 times a week for 12 weeks in a residential school	Executive function	In executive function tests, a significant improvement was reported after yoga; however, no change was observed in CG.
**Verma, Shete, Thakur, Kulkarni and Bhogal, 2014** [88]	Randomised controlled trial	YES—regular physical training	*n* = 82 11–15 years (mean 13.02) No gender details	‘Kaivalydhama Yoga’—Postures, breathing, chanting	Memory	The yoga group showed a significant improvement in the memory of the figural information test. The remaining memory tests did not show any improvements or changes. The CG showed no improvements in any of the memory tests.
**Vhavle, Rao and Manjunath, 2019** [89]	Randomised controlled trial	YES—traditional PE	*n* = 802 No age or gender details	Yoga—no intervention details 60 min × 5 days a week for 2 months	Executive function	Within groups, both CG and yoga significantly increased numerical executive function tests; however, there was no significant difference between groups. There was a significant increase in alphabetical executive function tests in the yoga group but not CG.
**Wang and Hagins, 2016** [96]	Qualitative	NO	*n* = 74 (approx) 9–12 years No gender details	Yoga—Postures, breathing, meditation, relaxation and didactic/themed discussions 1/2 × a week across a full academic year	Self-esteem, anxiety, academic performance	Increased self-esteem was one of the main findings reported by focus groups. Increased academic performance was also noted by some students.

**Table 5 children-09-00849-t005:** Summary of Neurodiverse and Neurotypical Findings.

Outcome	Neurodiverse	Neurotypical
Anxiety	**0/1**	**7/10**
Depression	**0**	**7/8**
Self-esteem	**0**	**7/7**
Self-concept	**2/2**	**1/2**
Psychological & Subjective Well-being	**2/2**	**12/15**
Resilience	**0**	**2/3**
Executive Function	**1/1**	**5/7**
Inhibition	**0**	**2/3**
Working Memory	**0**	**1/2**
Shifting	**0**	**0**
Attention	**1/2**	**15/15**
Academic performance	**3/3**	**12/13**

## Data Availability

Details of reviewed articles can be found in Table 3 and Table 4. The search terms and strategy can be found in the Appendix A along with the PRISMA Scoping Review checklist.

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
