# Peer review of "Scoping Review of Yoga in Schools: Mental Health and Cognitive Outcomes in Both Neurotypical and Neurodiverse Youth Populations"

_children, 2022, doi:10.3390/children9060849_

Round 1

Reviewer 1 Report

Dear authors,

First of all, thank you very much for taking into account my opinions when reviewing this article, which I hope will contribute to improving its quality.

With respect to the theoretical framework, I consider that it is correctly justified, which is why I consider that formatting issues should be reviewed, but it is not necessary to provide any more justification to clarify it.

With regard to the method, I would like a justification to be made that would allow us to understand why certain databases were searched and Scopus or WOS, for example, were excluded when performing the search and others were used. Please justify the selection of the databases.

When describing the results, it is vital that, next to each of the percentages, you include the citations of the authors to whom the study refers so that if you want to consult them, the reader can immediately take charge of which article you are referring to.
In my opinion, the tables that are presented in the results are very extensive and uncomfortable to read. From my point of view, it is more efficient to organize the information either written in results or simplified in the tables, since they are extremely uncomfortable to read.

I consider it interesting to contribute a little more scientific literature not present in the theoretical framework within the discussion.

Reviewer 2 Report

Dear Authors,

in my opinion, the topic is interesting considering the post COVID19 pandemic scenario and its psychological and social consequences on younger individuals. Yoga might represent an inclusive physical activity improving both physical and mental well-being in younger subjects.

The paper is well written and easy to read, it follows optimal study guidelines, and the results are intriguing.

However, some minor revisions are needed to improve the manuscript.

Revisions

WHOLE MANUSCRIPT. Please, correct page numbering because every time the format of the page changes the numeration starts over.

ABSTRACT. Page 1, line 16. Please, substitute the term “neurodivergent” with “neurodiverse” to be consistent with the main text.

INTRODUCTION. Page 2, line 42. Please, correct “A population that are higher risk for mental health comorbidities are neurodiverse” with “A population that is at higher risk for mental health comorbidities is neurodiverse”.

INTRODUCTION. Page 3, line 91. Please, correct “Research Question; What” with “Research question: what”.

RESULTS. Please, cite the appropriate reference every time you mention a study’s characteristics.

RESULTS. Page 7, line 236. Please, correct “and18%” with “and 18%”.

DISCUSSION. Page 30, line 496. Please, use capital letters for the subheading “Comparison with literature and plausible explanations for findings”, to be consistent with the other ones.

REFERENCE LIST. Please, according to the Instructions for Authors of the Journal, references should be reported as it follows: “1. Author 1, A.B.; Author 2, C.D. Title of the article. Abbreviated Journal Name YearVolume, page range.”.
